# Repeated Exposure to Subinfectious Doses of SARS-CoV-2 May Promote T Cell Immunity and Protection against Severe COVID-19

**DOI:** 10.3390/v13060961

**Published:** 2021-05-22

**Authors:** Maria Laura De Angelis, Federica Francescangeli, Rachele Rossi, Alessandro Giuliani, Ruggero De Maria, Ann Zeuner

**Affiliations:** 1Department of Oncology and Molecular Medicine, Istituto Superiore di Sanità (National Institute of Health), 00161 Rome, Italy; marialaura.deangelis@iss.it (M.L.D.A.); federica.francescangeli@iss.it (F.F.); rachele.rossi@iss.it (R.R.); 2Environment and Health Department, Istituto Superiore di Sanità, 00161 Rome, Italy; alessandro.giuliani@iss.it; 3Department of Medicine and Translational Surgery, Università Cattolica del Sacro Cuore, 00168 Rome, Italy; ruggero.demaria@unicatt.it; 4Institute of General Pathology, Policlinico Universitario Fondazione A. Gemelli, 00168 Rome, Italy

**Keywords:** COVID-19, SARS-CoV-2, protective immunity, T cell responses, facial masking, fomites, environmental exposure, memory T cells

## Abstract

Europe is experiencing a third wave of COVID-19 due to the spread of highly transmissible SARS-CoV-2 variants. A number of positive and negative factors constantly shape the rates of COVID-19 infections, hospitalization, and mortality. Among these factors, the rise in increasingly transmissible variants on one side and the effect of vaccinations on the other side create a picture deeply different from that of the first pandemic wave. Starting from the observation that in several European countries the number of COVID-19 infections in the second and third pandemic wave increased without a proportional rise in disease severity and mortality, we hypothesize the existence of an additional factor influencing SARS-CoV-2 dynamics. This factor consists of an immune defence against severe COVID-19, provided by SARS-CoV-2-specific T cells progressively developing upon natural exposure to low virus doses present in populated environments. As suggested by recent studies, low-dose viral particles entering the respiratory and intestinal tracts may be able to induce T cell memory in the absence of inflammation, potentially resulting in different degrees of immunization. In this scenario, non-pharmaceutical interventions would play a double role, one in the short term by reducing the detrimental spreading of SARS-CoV-2 particles, and one in the long term by allowing the development of a widespread (although heterogeneous and uncontrollable) form of immune protection.

## 1. Introduction

COVID-19 infection rates have resurged in most European countries after the 2020 summer period, giving rise to a tightly connected second and third wave of the pandemic. Despite the overall concern for the spread of the B.1.1.7 variant, which has been reported to cause a more severe illness as compared to previous SARS-CoV-2 viruses [1,2,3], COVID-19-associated mortality has not increased in parallel to the new surge of infections. Additionally, some studies performed in Italy and Spain observed a lower pathogenicity of COVID-19 in the second epidemic wave as compared to the first wave [4,5]. Several factors may have contributed to the current relative decrease in COVID-19 infection to mortality ratio as compared to the first wave. The optimization of clinical protocols for the management of severe COVID-19 cases, together with the availability of new therapeutic tools, largely contributed to the decline of death rates. An additional factor potentially affecting the decrease in COVID-19 fatalities is the younger mean age of affected subjects in the second pandemic wave. However, an analysis of German data showed that the fatality rate from COVID-19 has declined in all age groups, and that the older age groups drive the overall reduction [6], indicating that the “young patient prevalence” cannot be considered a universal explanation. A factor potentially confounding evaluations of epidemic trends is the fact that the number of tests has increased by several orders of magnitude between the first and the second/third pandemic wave, extending to unselected populations and amplifying case detection. However, in European countries the constant increase in screening rates is not coupled to the trend of positive cases [7] indicating that the number of positive cases does not depend only on testing efficiency. In this opinion, we explore the hypothesis that low doses of SARS-CoV-2 present in populated environments may induce virus-specific T cell responses, leading to progressive protective immunity. The hypothesis that low virus doses leaking from facial masks may reduce the severity of disease among people who become subsequently infected with SARS-CoV-2 was previously proposed in order to explain the observed increase in asymptomatic infections in settings with universal facial masking [8,9,10]. We extend this hypothesis by considering repeated contacts with viral particles suspended in the air, or present on inanimate objects as a potential source of T cell immunity and discuss this concept in the light of the latest immunological findings showing the presence of SARS-CoV-2 immunity in non-infected subjects.

## 2. Uncoupled Trends of COVID-19 Infection and Mortality Rates in European Countries, with a Focus on Italy’s Evolution of Disease Severity

According to the European Center for Disease Prevention and Control (ECDC) updated surveillance reports, in 2021, the majority of European countries experienced high rates of COVID-19 infection without a proportional increase in mortality rates [7]. Countries in the Europe geographical area at the time of manuscript submission could be roughly divided into two groups, one with increasing COVID-19 cases and decreasing/constant deaths (Georgia, Lithuania, Latvia, and Denmark), and one with both decreasing rates of infections and decreasing deaths (the remaining 51 countries). Italy belongs to the second group, with COVID-19 infection rates in decrease but still 3–4 times higher as compared to the first outbreak of March–April 2020 [7]. An up-to-date analysis, reporting the distribution of COVID-19 severity in Italy from February 2020 to April 2021, clearly shows a neat increase in the percentage of paucisymptomatic and asymptomatic cases, starting from the end of the first epidemic cycle (Figure 1) [11]. A detailed look at the trends of disease severity and mortality indicates that during the first COVID-19 wave, there was a clear shared trend as for the number of intensive care unit (ICU) accessions and the number of deaths [12]. Starting from October 2020, the dynamics of these indexes was decoupled from the cumulative number of deaths no longer following the number of ICU accessions. Consistent with previously shown dynamics of the relation between intensive care accessions and fatalities [12], the coupled increase of different severity classes ends at the beginning of summer 2020. From that time onward, the paucisymptomatic and asymptomatic cases neatly outnumber the more severe cases. The convergence of two completely independent analyses focusing on different endpoints is a signature of the robustness of the studied phenomenon. It is worth noting that the analysis shown in Figure 1 deals with surely ascertained cases, eliminating possible biases due to different definitions of ‘cause of death’. Moreover, the use of percentages instead of absolute numbers does eliminate the biases coming from the number of tests. The paucisymptomatic trend is probably the most informative, being less affected than the asymptomatic one to the presence of false positives to RT-PCR testing strategy. Overall, we propose that the current epidemiological situation in Italy (which mirrors Europe’s global situation, as reported by the ECDC) reflects on one side the spread of more transmissible SARS-CoV-2 variants (responsible for the increase in infections), and on the other side the progressive development of natural immunity among the population (resulting in a large prevalence of asymptomatic and paucisymptomatic cases). As discussed in the following sections, the severe lockdown measures imposed by several European countries during the first pandemic wave likely resulted in decreased amounts of circulating virus, possibly influencing subsequent trends of disease severity. The current epidemiological picture in European countries is in contrast with that of several Latin American countries (i.e., Brazil, Mexico, Peru, Paraguay), where recurrent waves of COVID-19 were coupled to a parallel increase in mortality rates [7]. Although multiple and complex factors contribute to influence pandemic parameters, it may be inferred that at least some countries that did not apply effective public health measures during the first COVID-19 wave may have experienced more severe disease manifestations thereafter. Finally, several considerations suggest that in the long-term, SARS-CoV-2 will lose its potency and become endemic, similarly to common cold coronaviruses (CCCoVs) [13,14]. This process depends on multiple factors, including the duration of protective immunity, susceptibility to reinfection, seasonality, competition with other circulating respiratory viruses and control measures, as analyzed in detail elsewhere [13]. In this view, it may be hypothesized that the variable degrees of immunity, possibly elicited by low-dose SARS-CoV-2 particles present in the environment, may favor a softer transition of SARS-CoV-2 to an endemic state, by contributing to a progressive decrease of COVID-19 severity.

## 3. Possible Role of Low Virus Doses in Eliciting a Protective Immune Response against SARS-CoV-2

The concept that low viral exposure can lead to milder disease manifestations followed by immunization provided the roots for the birth of vaccination procedures and has been investigated in multiple experimental settings, ranging from parasites to different viruses [15,16,17,18,19,20]. The use of low antigen doses is gaining increasing attention also in vaccine development, as several studies pointed to a higher quality of T cell responses obtained through low levels of antigenic stimulation [21]. The observation that low virus doses result in milder disease manifestations has found confirmation in mouse-adapted models of MERS and SARS [22,23], in a hamster model of SARS-CoV-2 infection [24] and recently in ferrets infected with SARS-CoV-2 [25]. Low dose challenge models for SARS-CoV-2 have not yet been systematically developed [26]. A major impediment to this approach is that low dose inoculation may lead to only a portion of animals being infected, thus requiring large numbers of animals to achieve statistical significance. Nevertheless, some studies have explored low dose challenge with SARS-CoV-2, using nebulized virus, or co-housing of infected and uninfected animals [27,28,29,30]. Interestingly, human challenge models of SARS-CoV-2 infection have also been proposed [31,32,33]. A fieldwork in humans supporting the statement that low SARS-CoV-2 doses induce less severe disease manifestations as compared to high doses is provided by studies performed on front-line healthcare workers. This category has been previously shown to be at high risk for COVID-19 infection [34]. Recent studies on 2884 exposed healthcare workers showed that insufficient access to personal protective equipment was associated with increased likeliness of COVID-19 infection, increased disease severity and prolonged duration, supporting the hypothesis that SARS-CoV-2 viral inoculum could be associated with disease severity [35]. Low doses of viral particles leaking through surgical masks have been proposed to act as a sort of “variolation” process, increasing the proportion of asymptomatic SARS-CoV-2 infections and leading to natural immunity [9,10]. This hypothesis seems confirmed by several studies showing decreased COVID-19 infections and disease severity upon mask wearing [35,36,37]. Here, we would like to take into consideration two additional factors potentially contributing to the development of natural immunity to SARS-CoV-2 in part of the population. The first factor is represented by the presence of the virus on collective objects, such as money, handrails in public transportation and elevators, cash machines, intercoms and doorknobs. Previous studies on SARS-CoV-1 have reported the presence of coronavirus on fomites (inanimate objects) [38]. Moreover, contamination of environmental surfaces has been proposed to explain unusual clusters of infection by SARS-CoV-1 [39,40]. The presence of SARS-CoV-2 on fomites has been reported to occur both in hospitals and in non-hospital settings [41,42,43]. The environmental contamination with SARS-CoV-2 has been previously proposed to be scarcely relevant for the spread of infection throughout the population [44]. Recently, increased understanding of SARS-CoV-2 presence and stability on fomites led the Centers for Disease Control and Prevention (CDC) to state that the relative risk of fomite transmission of SARS-CoV-2 is low, compared with direct contact, droplet transmission, or airborne transmission, as stated by a continuously updated report on this topic [45]. In fact, systematic analyses of virus stability on fomites indicated that the infectious titer of SARS-CoV-2 on plastic or steel was greatly reduced after 72 h [46]. Moreover, several studies that analyzed the infectivity of swabbed samples concluded that the virus present on surfaces was not able to infect Vero E6 cells, presumably due to the low dose of viral inoculum and to the decrease of virus viability [38,47]. This said, we postulate that low doses of environmental SARS-CoV-2 may still be important in eliciting protective immunity against COVID-19, due to the recurrent and multiple nature of antigenic contacts. In addition to viral particles that enter the lungs through breathing, low doses of the virus may also reach the gastrointestinal system through the enteral route, i.e., by touching buccal mucosa or food with contaminated hands. In fact, hand washing (which is obviously not a sterilization procedure) does not totally eliminate viral particles, as SARS-CoV-2 is not completely inactivated by normal detergents [46]. In this case the uptake of small virus amounts may occur in the gut, where epithelial cells express ACE2 and can be efficiently infected by SARS-CoV-2 [48]. SARS-CoV-2 exposure in the gastrointestinal tract has been previously shown to be less pathogenic as compared to the respiratory tract [49] and has been proposed to contribute to COVID-19 asymptomatic infections [32]. A factor that may crucially influence the generation of protective immunity upon low antigenic stimulation may be represented by the repeated contacts of the immune system with the pathogen, according to the principle of recall widely exploited in vaccination schedules. Recurrent antigenic stimulation may also include contacts with a mixture of infectious and non-infectious viral particles. In fact, although infectious SARS-CoV-2 particles present on environmental surfaces decrease with time, viral antigens exposed on non-infectious viruses may still be useful to alert the immune system. According to this principle, exposure to inactivated pathogens is exploited by several vaccines, such as those against hepatitis A, rabies, flu, and polio. Inactivated pathogens are known to be less effective than live germs in eliciting protective immunity, and therefore vaccines with inactivated pathogens often require several doses over time in order to develop efficient immunization. Interestingly, whole microorganism vaccines (bacille Calmette–Guérin/BCG, oral polio vaccine, measles) are known to induce a long-term boosting of innate immune responses termed trained immunity, which is able to induce heterologous protection against other infections through the reprogramming of innate immune cells. Trained immunity by whole microorganism vaccines has been proposed to reduce SARS-CoV-2 susceptibility and severity [50,51]. Several trials are underway to determine whether vaccination with BCG can help prevent Covid-19 and will help to clarify this issue. Due to the very low (although detectable) doses of SARS-CoV-2 present in the environment, a repeated contact with viral antigens may be essential for the development of specific immune responses. In addition to the multiplicity of antigenic contacts, immunization by low virus doses is likely influenced by a number of other factors. Among these factors, the most important is likely the fitness of the host immune systems, which will respond to antigenic challenges according to the individual’s health, age, and immunological competence. Moreover, previous exposure to CCCoVs is emerging as an important factor, potentially influencing the immune response against SARS-CoV-2. In fact, several studies have identified T cells reactive against SARS-CoV-2 antigens in a relevant percentage (20-50%) of unexposed donors, i.e., pre-pandemic blood samples [52,53,54,55,56]. Lymphocytes from pre-pandemic samples that react against SARS-CoV-2 antigens likely represent memory T cells generated upon previous exposure to endemic CCCoVs, such as HCoV-OC43, HCoV-HKU1, HCoV-NL63 and HCoV-229 [57]. At present, it is under debate whether such cross-reactive T cells are actually effective in providing protection against COVID-19. Some studies indicate that a recent documented CCCoVs infection is associated with less severe COVID-19, suggesting the involvement of lung-localized memory T cells and B cells in protecting from severe disease manifestations [58]. By contrast, other studies have not detected an association between previous CCCoVs infection and COVID-19 severity [59], or even reported a worse clinical outcome in COVID-19 patients with CCCoV humoral immunity [60]. Interestingly, memory CD4+ and CD8+ T cells, cross-reactive against SARS-CoV-2 detected in unexposed healthy donors, failed to expand in vitro, which suggests they have limited potential to function as part of a protective immune response against COVID-19 [61]. In addition to pre-existing T cell immunity, recent studies showed that approximately 20% of pre-pandemic healthy donors possessed SARS-CoV-2 cross-reactive serum antibodies, but these antibodies were not associated with protection from SARS-CoV-2 infections, or disease severity [62].

## 4. Do Repeated Exposures to Low SARS-CoV-2 Doses induce T Cell Immunity? A Possible Explanation for SARS-CoV-2-Specific T Cell Responses in Non-Infected Subjects

Recent studies on immunological responses against SARS-CoV-2 provided a detailed picture of immune signatures in COVID-19 patients (either with severe disease, mildly symptomatic or asymptomatic) as compared to convalescent individuals, people that have been in close contact with a COVID-19 patient, and unexposed healthy donors. Such studies consistently reported a major role of T cell responses in developing immunity against SARS-CoV-2 [53,55,63,64,65,66,67,68,69]. Notably, T cell responses were shown to be polyfunctional (involving both CD4+ and CD8+ lymphocytes), stable, and lasting for at least six months after infection, even though anti-SARS-CoV-2 antibodies decline with time [63,65,69]. Moreover, SARS-CoV-2-specific T cells with a stem-like memory phenotype (CCR7+CD127+CD45RA−/+TCF1+) were identified in COVID-19 convalescent subjects, even in the absence of detectable circulating antibodies, highlighting the role of memory T cells in sustaining long-term immunity against SARS-CoV-2 [67]. A key study that quantified SARS-CoV-2-specific memory T cell responses across distinct cohorts of Swedish subjects confirmed the presence of pre-existing immunity against SARS-CoV-2 in unexposed donors, but also highlighted unprecedented details in this regard. Specifically, the authors found T cell responses directed against SARS-CoV-2 spike and/or membrane proteins in 28% of healthy individuals who donated blood before the pandemic, in 46% of healthy individuals who donated blood during the pandemic, in 67% of exposed family members (who shared a household with a COVID-19 patient at the time of illness), in 87% of convalescent subjects with a history of mild COVID-19, and in 100% of convalescent individuals recovering from severe COVID-19 [67]. While the presence of T cells reactive against SARS-CoV-2 in the blood of pre-pandemic donors is explained by previous infection with CCCoVs, the nearly doubled percentage of SARS-CoV-2-reactive T cells in subjects who donated blood after the beginning of the pandemic strongly suggests that contact with viral particles present in the environment has elicited specific T cell responses. Moreover, the striking increase of SARS-CoV-2-reactive T cells in exposed family members (one third of whom were antibody-seronegative) further supports the hypothesis that a low-dose viral inoculum, presumably reiterated over time, resulted in the development of specific T cell responses. This hypothesis gains further support from recent studies showing that individuals who stayed in close contact with a COVID-19-positive subject developed T cell memory in the absence of infection [61]. Close contacts are often negative to both nucleic acid testing and antibodies, indicating that SARS-CoV-2 failed to establish a successful infection [61]. A similar observation was reported during the MERS epidemic, where high-risk individuals (camel workers) who were nucleic acid testing negative and antibody negative developed significant levels of MERS-CoV specific memory T cells [70]. It remains to be determined whether T cell responses generated in the absence of infection can protect against subsequent challenges with SARS-CoV-2. However, previous studies on influenza-specific T cells demonstrated that pre-existing memory CD4+ and CD8+ lymphocytes were able to provide protection against subsequent infections [71,72,73]. Also in the case of coronaviruses, specific memory T cells have been shown to provide protective immunity against severe infection [74,75,76], thus providing support to the hypothesis that SARS-CoV-2-specific T cells developed upon natural exposure may prevent recurrent episodes of severe COVID-19. In summary, the presence of T cells and antibodies against SARS-CoV-2 epitopes derived from previous CCCoVs infection has been consistently detected in unexposed individuals, although the role of such pre-existing immunity in protecting from COVID-19 is not clear. Pre-existing immunity derived from CCCoVs cross-reactivity does not exclude the possibility that SARS-CoV-2-specific T cells may develop upon contact with sub-infectious virus doses. The latter immune response, although less efficient as compared to that developed by COVID-19-infected individuals, is likely more specific as compared to anti-CCCoVs cross-reactive immunity and may play a role in modulating the susceptibility and severity of SARS-CoV-2 infection. Finally, an important question to consider is the recent emergence of SARS-CoV-2 variants and whether pre-existing immune responses to provide at least partial protection against mutated viral particles. While awaiting specific studies to address this topic, the observation that immune responses develop against multiple different regions of viral proteins suggests that previous contacts with SARS-CoV-2 may favor population immunity against reinfection [14]. In this scenario, limiting SARS-CoV-2 circulation through non-pharmaceutical interventions and the scale-up of vaccines will restrain viral replication and consequently limit the generation of new variants until population immunity is achieved. 

## 5. Possible Differences in the Immune Response against High and Low SARS-CoV-2 Doses

An explanation of the hypothesis that repeated antigenic stimulation with low SARS-CoV-2 doses can elicit protective immunity may lay in the difference between the immune mechanisms activated upon the encounter with high versus low doses of the virus (Figure 2). High doses of SARS-CoV-2 reaching the pulmonary alveoli are more likely to result in enhanced kinetics of viral replication. Large amounts of viral particles can then induce the death of airway epithelial cells with consequent release of damage-associated molecular patterns (DAMPs). DAMPs trigger the production of inflammatory cytokines and chemokines that recruit monocytes, neutrophils, and T cells to the lungs. An important role in dictating the outcome of immune response to SARS-CoV-2 is played by mononuclear phagocytes (MNP) present in the bronchoalveolar microenvironment. In fact, single cell RNA-sequencing of bronchoalveolar fluid from patients with severe or mild COVID-19 showed that in severe cases, the MNP population was characterized by a depletion of tissue-resident alveolar macrophages (FABP4+) and by an abundance of inflammatory monocyte-derived macrophages (FCN1+SPP1+) [77]. Inflammatory macrophages are responsible for chemotaxis, for propagating inflammation and for the generation of an immune-suppressive milieu in SARS-CoV-2-infected tissues [78]. In the severe phase of COVID-19, lung inflammation leads to diffuse alveolar damage and acute respiratory distress syndrome, possibly accompanied by systemic inflammation, widespread coagulopathy and multiorgan dysfunction. In this scenario, the immune system undergoes profound alterations, including a dysregulated activation of monocytes and neutrophils and a decrease in natural killer cells and peripheral blood T cells [78]. By contrast, low doses of SARS-CoV-2 reaching the pulmonary alveoli are less likely to trigger massive viral replication, unless the host T cell immunity is compromised. In the absence of inflammation, low doses of viral antigens come in contact with airway epithelial cells and alveolar macrophages, therefore being processed and presented, respectively, to CD4+ T cells through major histocompatibility complex (MHC) class II or to CD8+ T cells through MHC class I. This process may result in the development of an adaptive immune response of variable strength against SARS-CoV-2. In this regard, it is improbable that exposure to low SARS-CoV-2 doses would result in sterilizing immunity, but the presence of memory T cells in the lungs would minimize COVID-19 disease severity to that of a “common cold” or asymptomatic disease. In fact, SARS-CoV-2–specific CD4+ T cells and CD8+ T cells are associated with less COVID-19 disease severity during an ongoing SARS-CoV-2 infection [65,79]. Interestingly, CD4+ T cell response against SARS-CoV-2, which was found in individuals exposed to the virus but not infected, has been proposed to occur also in the absence of viral replication [61]. In fact, while a robust CD8+ response occurs in the presence of abundant viral antigens usually generated by viral replication, CD4+ response and subsequent immunological memory does not rely on endogenous viral replication, but involves endocytosis/phagocytosis of exogenous viral antigens, which are mostly derived from non-replicative viral particles or soluble viral proteins [61,80]. Therefore, the formation of CD4+ T cell memory may be more easily achieved in uninfected exposed individuals [61]. Reiterated exposures to sub-infectious virus doses may consolidate T cell responses giving rise to memory T cells responsible for efficient long-term immunization. Most likely, the generation of protective immunity upon exposure to low virus doses occurs in individuals able to mount a proficient immune response, as low amounts of antigen may fail to activate immune systems with suboptimal or compromised efficiency. The lack of protective immunization by previous exposure to low virus doses may contribute to explain why individuals with older age or concomitant pathologies experience more severe forms of COVID-19 [81]. In particular, older adults have been reported to have a decline in immune functions, a higher prevalence of chronic diseases and increased background levels of inflammation. These factors, along with the inability to develop an efficient immunological memory upon natural exposure to environmental SARS-CoV-2, could collectively account for more severe manifestations of COVID-19 in the older age group [82].

## 6. Conclusions

This opinion explores the hypothesis that naturally occurring exposures to low SARS-CoV-2 doses may promote the development of immunological memory, possibly contributing to protect against subsequent severe COVID-19 manifestations. This hypothesis is tightly linked to the temporary dampening of virus spreading imposed by national lockdowns. This effect may have allowed a higher number of low-dose exposures possibly resulting in various degrees of immunization of non-infected subjects, who would mount a more effective immune response upon a subsequent encounter with the virus. It is important to stress that, even in case this hypothesis is confirmed by further studies, it does not support a beneficial effect of SARS-CoV-2 exposure. In fact, the potential development of protective immunity upon exposure to low dose viral particles would depend on multiple and uncontrollable factors, such as the effective virus dose, the fitness of the host’s immune system, the presence of comorbidities/inflammatory states, and the influence of individual genetic factors (including several risk-associated genomic regions). Moreover, the spread of new SARS-CoV-2 variants in vulnerable populations could still lead to greater disease severity and mortality, supporting the continued implementation of protective measures and the rapid vaccination of at-risk individuals. Finally, if repeated low-dose exposures to SARS-CoV-2 are actually able to elicit T cell-mediated immunological memory, they would be useful in maintaining long-term anti-SARS-CoV-2 immunity post-vaccination. In fact, prolonged SARS-CoV-2 antigen persistence (although at low doses) might represent a continuous trigger for T cell responses, possibly expanding the antigenic repertoire to new circulating COVID-19 variants. In this view, non-pharmaceutical interventions, vaccinations, post-infection immunity, and possibly protection conferred by low environmental SARS-CoV-2, may altogether contribute to maintain post-pandemic immunity. 

## Figures and Tables

**Figure 1 viruses-13-00961-f001:**
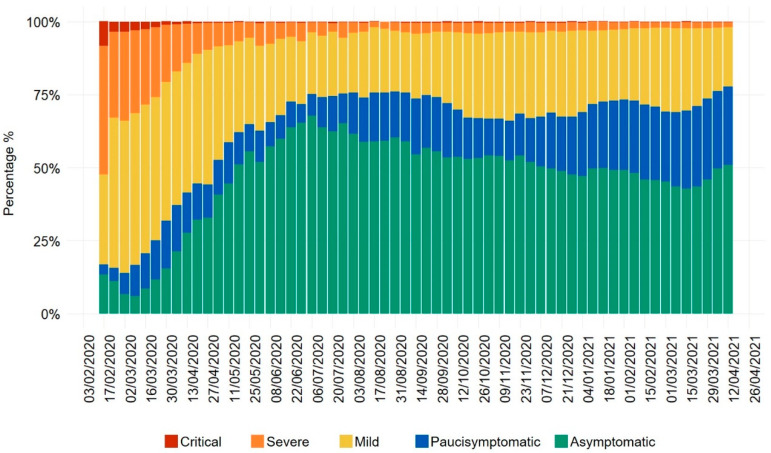
Epidemiological data of the Italian National Institute of Health (Istituto Superiore di Sanità, Rome, Italy) showing the clinical status of COVID-19 patients from January 2020 to April 2021. Colored bars represent the percentage of cases, respectively, with critical clinical conditions (top bars, dark red, clinical manifestations affecting the respiratory tract and/or other organ systems that require hospitalization in intensive care), with severe disease (orange bars, clinical manifestations affecting the respiratory tract and/or other organ systems that require hospitalization, not in intensive care), with mild disease (yellow bars, clinical manifestations affecting the respiratory tract and/or other organ systems that would not normally require hospitalization), with paucisymptomatic disease (blue bars, mild and general symptoms e.g., general malaise, fever, fatigue, etc.), and with asymptomatic disease (green bars, no apparent signs or symptoms of disease). Raw data in the xlsx format are publicly available at the Istituto Superiore di Sanità website www.epicentro.iss.it/coronavirus/sars-cov-2-dashboard (accessed on 23 April 2021).

**Figure 2 viruses-13-00961-f002:**
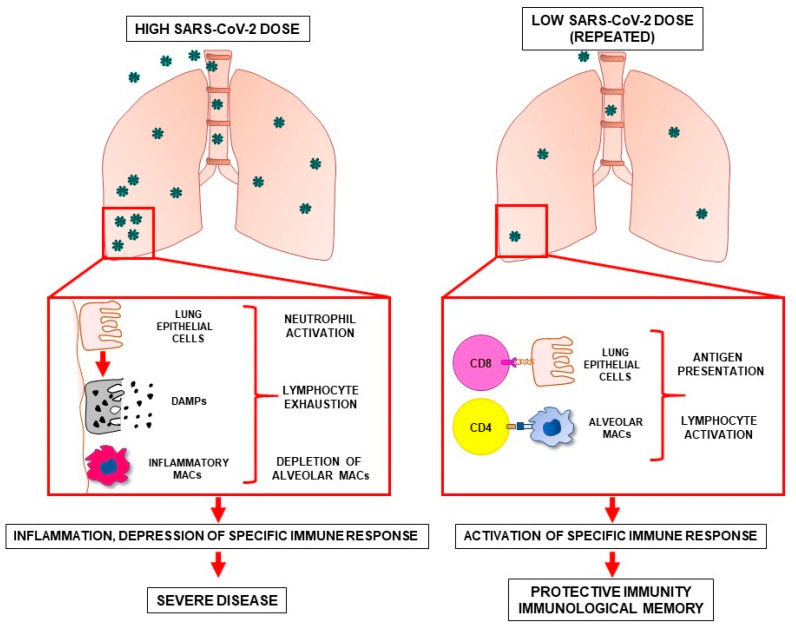
Modifications occurring in the lung microenvironment upon exposure to high doses (**left**) or low doses (**right**) of SARS-CoV-2. High (**left**) or low (**right**) numbers of SARS-CoV-2 particles enter pulmonary alveoli and infect airway epithelial cells. In the left panel, massively infected airway epithelial cells undergo cell death releasing damage-associated molecular patterns (DAMPs). DAMPs activate neighboring cells to produce inflammatory cytokines and recruit activated monocytes, macrophages, and neutrophils. In the inflamed lung microenvironment T lymphocytes, after an initial activation, undergo functional exhaustion, while inflammatory monocyte-derived macrophages (inflammatory MACs) substitute alveolar macrophages. In the right panel, low virus doses come in contact with airway epithelial cells and alveolar macrophages. Viral particles are processed and presented to T cells by lung epithelial cells and alveolar macrophages (alveolar MACs), thus generating a controlled immune response. Protective immunity may evolve into efficient long-term immunological memory upon repeated antigenic stimulation.

## Data Availability

Raw data reported in Figure 1 are publicly available in the xlsx format at the Istituto Superiore di Sanità website www.epicentro.iss.it/coronavirus/sars-cov-2-dashboard (accessed on 23 April 2021).

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
