# Peer review of "Repeated Exposure to Subinfectious Doses of SARS-CoV-2 May Promote T Cell Immunity and Protection against Severe COVID-19"

_viruses, 2021, doi:10.3390/v13060961_

Round 1

Reviewer 1 Report

De Angelis et al review data showing dramatically different SARS-CoV-2 case counts, hospitalizations, and deaths between early and later waves in Italy. They propose a hypothesis that the increased paucisymptomatic and asymptomatic cases in the later waves may be due to low level exposure to the virus (e.g., decreased by fomite or food exposure) that is generating improved T cell memory and less deadly responses in the later waves of infection. The review provides a possible explanation for observations made in Italy, but the review should be expanded to address the following deficiencies:

  • 1) While the data from Italy in figure 1 is valuable, it is important to expand the discussion to other countries around the world. How many countries are following this same trend and how many are not? How universal is this observation and why might some countries not be following this same trend?
  • 2) Many viruses evolve over time to be less virulent. Should address when this does and does not happen and whether reduced fitness or changes in the virus or variants could be the explanation for decreased death in the case of SARS-CoV-2.
  • 3) Should specifically define what is meant by paucisymptomatic in sentence 81 and figure 1. There is asymptomatic and mild disease, how is paucisymptomatic defined in your data analysis specifically.
  • 4) Typo’s is line 67 (“in the light” rather than “at the light”) and 72 (“are” instead of “is”).
  • 5) Should further clarify how your hypothesis of low levels of exposure (due to fomites or food exposure) is novel. It is not novel that we could be exposed via these routes, but maybe that T cell memory can be formed? Should provide more background on whether this idea has been proposed by others or if this is a new idea.
  • 6) Low level exposure via masks providing enhanced immunity has been proposed, this should be discussed more in depth.
  • 7) Should include enhanced discussion on the increased understanding on how little risk there is for being infected by virus on fomites: https://www.cdc.gov/coronavirus/2019-ncov/more/science-and-research/surface-transmission.html
  • 8) While addressing different mechanisms involved in the immune response against SARS-CoV-2, it would be important to address the potential role for trained immunity of the innate immune system and how this could impact T cell function and response.
  • 9) Countries have taken various approaches to lock downs, providing data that can provide insights into immunity. How does immunity in countries with and without strict lock downs compare in the same analysis done on the Italian population?
  • 10) The review would be stronger if specific experiments or data analysis that could be done to test the hypothesis that low level exposure via fomites or food is increasing T cell memory. How could this be tested in vitro, in vivo, and with the data we currently have.

Author Response

Reviewer 1

We are grateful to the Reviewer for his/her careful evaluation of the manuscript and for highlighting several points that needed explanations or corrections. We would like to underline the fact that the manuscript was erroneously submitted as a review, while its original intent was to be a viewpoint, or alternatively an opinion. We have asked the Editor to make this correction. This misunderstanding explains, at least in part, the rather hypothetical spirit of the manuscript and why our work did not include a systematic analysis of all countries.

1) We have added some considerations to other countries at page 3 of the revised manuscript. However, we did not further expand the discussion on this topic because we could not obtain detailed epidemiological data similar to those shown in Figure 1 for other countries.

2) We thank the Reviewer for the insight on the possible “endemic fate” of SARS-CoV-2, which is very relevant to our hypothesis. We have discussed the possible transition of SARS-CoV-2 towards an endemic state at page 3 of the revised manuscript, although we did not go deep into this topic as there are excellent comments on this subject (Shaman and Galanti, Science 2020; Veldhoen and Simas, Nature Reviews Immunology 2021), now cited in the revised manuscript. Concerning the question whether virus variants may explain decreased death rates, it has been reported that B.1.1.7 associates with increased severity and mortality. Therefore it cannot explain trends of decreasing COVID-19 severity observed in Italy and other European countries. We have added an additional reference on this topic (Grint DJ et al., Euro Surveill. 2021).

3) We have provided the explanation for all the terms referring to the different grades of COVID-19 severity (paucisymptomatic, asymptomatic, mild, severe and critical) in the Legend to Figure 1.

4) We have corrected the typos, thank you for the indications.

5) Previous studies (all cited in the revised manuscript) have proposed some aspects of the work presented here, and specifically:

- Gandhi and coworkers proposed that viral particles leaking from facial masks produced a sort of “variolation”.

- Hausdorff and coworkers proposed that low-dose and/or orally transmitted SARS-CoV-2 may be useful in COVID-19 prevention and envisaged controlled human infection studies to clarify this issue.

To our knowledge, however, our work explores for the first time the possibility that low dose SARS-CoV-2 specifically stimulates a T cell response in the respiratory or intestinal tracts in the absence of infection. Also, our work links this hypothesis with unpublished data on the Italian course of disease severity and discusses immunological observations seemingly supporting our hypothesis (in particular studies by Sekine at al., Cell 2020 and Wang et al., Nature Communications 2021). Finally, our work proposes for the first time that multiple exposures to low SARS-CoV-2 doses may be required for the progressive development of immunological memory in the absence of infection.

6) We have cited updated work from Gandhi’s group that reports available evidences on immunity provided by facial masks (Spinelli et al., Lancet Infect. Dis. 2021) and discussed the fact that personal protection devices (mainly masks) reduce COVID-19 infections and severity in health workers continuously exposed to viral challenges (Kim et al., BMJ Global Health 2021).

7) We added additional reference to the fact that fomites are associated with low risk of infection with SARS-CoV-2 and cited the CDC web page that presents updated evidence on this topic.

8) We agree with the Reviewer that trained immunity may impact on T cell function and response and added a reference to this topic in the revised manuscript.

9) We have introduced a brief comparison of COVID-19 severity between Italy (and other European countries that implemented early lockdowns) and countries where public health measures were not present, especially in South America.

10) In the revised version of the manuscript we have reported additional studies reporting the association between low dose SARS-CoV-2 inoculum and disease severity in animal models (Ryan KA et al., Nature Communications 2021) and cited several studies that proposed to test the “low-dose hypothesis” in controlled human studies (Hausdorff VP et al., Int. J. Infect. Dis. 2021; Deming ME et al., New England J. Med 2020; Nguyen LC et al., Clin. Infect. Dis. 2021). In addition, the recent paper by Spinelli et al. (Spinelli MA et al., The Lancet Infect. Dis. 2021) explores the correlation between low dose inoculum and COVID-19 transmission or severity in human settings. Moreover, the recently published study by Wang et al. (Nature Communications 2021) shows that a substantial fraction of individuals in close contact with COVID-19 patients develop immunity in the absence of infection, providing support to our hypothesis.

Author Response

Reviewer 2

We are grateful to the Reviewer for his/her accurate analysis of our manuscript and for his/her suggestions, which allowed us to significantly improve the quality of our work. We would like to underline the fact that the manuscript was erroneously submitted as a review, while its original intent was to be a viewpoint (or alternatively an opinion). This fact explains the relatively low number of references and the rather hypothetical spirit of the whole work. However, we agree with the Reviewer that the hypothesis presented in the manuscript required further substantiation, which was provided in the revised version. Among the amendments made to the manuscript we recognized the fact that cross-reactivity with common cold coronaviruses (CCCoVs) was widely proven to be responsible for pre-existing immunity against SARS-CoV-2. This observation, as correctly stated by the Reviewer, does not exclude a possible role of a specific SARS-CoV-2 immune response developed in the absence of infection.

Major comments

1) We have listed new references referring to experimental models using low doses of viruses (including MERS, SARS-CoV and SARS-CoV-2) and to the observation that low dose inocula result in decreased disease severity. The idea that low SARS-CoV-2 doses may provide immunization is a hypothesis that stems from the following observations:

a) the relationship between low dose inoculum and reduced disease severity,

b) immunological studies showing that mild infections, asymptomatic infections or even contact with COVID-19 positive individuals in the absence of infection (recent study by Wang et al, doi 10.1038/s41467-021-22036-z) result in the generation of a specific immune response,

c) epidemiological studies showing that masks use seems linked to less infections and decreased disease severity,

d) the trends of decreasing disease severity over time observed in Italy and other countries that implemented public health protective measures as compared to countries where protective measures weren’t applied,

e) previous studies on influenza-specific T cells demonstrated that pre-existing memory CD4+ and CD8+ lymphocytes were able to provide protection against subsequent infections. Also in the case of coronaviruses, specific T cells have been shown to provide protective immunity against severe infection.

However, we acknowledge the fact that there are yet no experimental evidences proving that contact with low virus doses may result in immunization or decreased disease severity and we have highlighted this in the revised version of the manuscript.

We have clarified the difference between pre-existing T cell immunity (due to crossreactivity with CCCoVs epitopes) and SARS-CoV-2 specific T cell responses.

2) We agree with the Reviewer that crossreactivity between SARS-CoV-2 and CCCoVs epitopes has been recognized by numerous studies and we acknowledged this fact in the revised manuscript. We also mentioned the fact that the presence of pre-existing immunity against CCCoVs does not exclude the subsequent development of specific immunity against SARS-CoV-2 elicited by subinfectious virus doses.

3) We have listed new references referring to experimental models using low doses of viruses (including MERS, SARS-CoV and SARS-CoV-2) including the excellent work by Khoury et al. suggested by the Reviewer. We have further stressed the hypothetical nature of our work and removed or modified sentences that may be erroneously interpreted as an encouragement to expose to SARS-CoV-2. In addition, we added the following sentence to the Conclusions: “It is important to stress that, even in case this hypothesis would be confirmed by further studies, it does not support a beneficial effect of SARS-CoV-2 exposure. In fact, the potential development of protective immunity upon exposure to low dose viral particles would depend on multiple and uncontrollable factors such as the effective virus dose, the fitness of the host’s immune system, the presence of comorbidities/inflammatory states and the influence of individual genetic factors (including several risk-associated genomic regions)”.

4) We provided a clearer explanation of the fact that only very low doses of SARS-CoV-2 may be useful to the development of memory T cells, while high doses (that are present in the absence of hand washing, sanitization, mask wearing and restrictions) can be detrimental.

5) We have removed the ambiguous sentence.

6) We have provided two references on COVID-19 in older adults, one of which provides a rationale for the fact that older people experience a more severe disease (British Society for Immunology, 10 November 2020 “The ageing immune system and COVID-19”). We have also added a sentence in the text to summarize these findings (“In particular, older adults have been reported to have a decline in immune functions, a higher prevalence of chronic diseases and increased background levels of inflammation, which could collectively account for more severe manifestations of COVID-19 in this age group”).

Minor comments

1) We apologize for the inconvenience that occurred to the bibliography.

2) We have used the terms “unexposed” and “non-infected” (or “uninfected”) more appropriately throughout the text.

3) We have modified this sentence in the Conclusions as it was ambiguous.

Reviewer 3 Report

De Angelis and colleague discuss here their hypothesis that repeated SARS-CoV-2 exposure explains specific T cell responses in unexposed individuals. I find it an interesting read that felt however more like an opinion rather than a review. Nevertheless, I acknowledge the value of the text but I would strongly advise balancing the argumentation.

The title is misleading: “Repeated exposure (…) in unexposed individuals.” An individual does not need to develop the disease to be considered exposed. I would argue that people with repeated exposure at subinfectious doses are in fact exposed individuals. Unexposed is a termed that should be reserved for people that have never seen SARS-CoV-2 antigen, like in the pre-pandemic era.

It would be useful to list the countries in the three groups mentioned.

The following sentence is highly speculative: “As described in the following sections, the total lockdown imposed by several European countries during the first pandemic wave may have contributed to lower the amount of circulating virus, thus favoring the prevalence of immunizing exposures over infective ones.” I follow the authors’ reasoning, but their statement is an oversimplification. Some levels of viruses must be circulating otherwise there would be no emergence of immunity at all. There is a delicate balance between too much and too few. The text should be edited to reflect this nuance.

Healthcare workers are routinely exposed to SARS-CoV-2, especially respiratory therapists. It would be interesting to include such specific examples in the manuscript, with reference to document. Many publications include cohorts of healthcare workers.

I strongly disagree with this statement: “However, bioinformatic studies showed that the pre-formed SARS-CoV-2 immunity is unlikely due to cross-reactivity with other coronavirus epitopes (34)”. Many studies reported cross-reactivity with other coronaviruses by testing pre-COVID-19 samples (Shrock et al. 2020 in Science; Le Bert et al. 2020 in Nature; Mateus et al. 2020 in Science). Repeated low dose exposure with SARS-CoV-2 certainly cannot explain pre-existing immunity in these cases.

The argumentation about the emergence of anti-SARS-CoV-2 immunity should be nuanced. True, a higher infectious dose can cause more inflammation. But this unlikely due to the direct toxicity of this initial dose but rather by accelerating the kinetics of replication. Even a high infectious dose is nothing compared to the amount of virus shed per minute of the viral amplification phase. A faster spread can take the innate immunity by surprise, forcing it to overcompensate by increasing inflammation in a way that may quickly become out of control and detrimental. It is particularly true for CD8 + T cells because they classically require epitope presentation via MHC-I generally from infected cells. Please clarify the role of viral replication in the inflammatory cascade.

Innate responses take place before humoral responses. At a low dose, the infection is likely cleared before the humoral response can be generated. In this regard, a fine equilibrium is necessary to elicit a protective immunity: too low and innate immunity suffice, too high and the innate immunity causes out-of-control inflammation. In between, the most frequent occurrence, the innate immunity buys some time so humoral responses can emerge.

I like how they rationalize the importance of reducing the input of infectious viruses via non-pharmaceutical interventions.

Cross-reactive immunity is expectedly of low yield and may not be detected with current assays. Repeated low dose exposures are likely to amplify this immunity, which will likely become detectable. Likewise, we could expect that repeated low dose exposures will help maintain a long-term anti-SARS-CoV-2 immunity post-vaccination. This could be developed in the manuscript.

Author Response

Reviewer 3

We are grateful to the Reviewer for his/her thorough evaluation of the manuscript and for his/her helpful suggestions. We agree with the observation that the manuscript would make more sense as an Opinion rather than a Review: in fact, it was erroneously submitted as a Review, while in its original intent it was intended as a Viewpoint (or alternatively an Opinion). We have asked the Editor to correct its categorization.

  • We have corrected the confusing use of the terms “exposed” and “unexposed” and changed the title of the manuscript to a clearer form.
  • Since the first manuscript submission, the situation of European countries in relation to COVID-19 infection and deaths has radically changed: the group with increasing infections and deaths disappeared, the group with increasing infections and decreasing deaths now includes only five countries (Belarus, Georgia, Lithuania, Latvia and Denmark) and the third group (decreasing infections and deaths) includes the remaining 51 countries, as listed by the CDC website ((https://covid19-country-overviews.ecdc.europa.eu/#2_Global). We have modified the manuscript text to acknowledge these changes but we thought it wasn’t worth to add a table as the situation is almost homogeneous between the majority of countries.
  • We have modified this part of the text.
  • We thank the Reviewer for this insight, which is very relevant to our hypothesis. Accordingly, we mentioned the case of healthcare workers in the revised manuscript and we added the reference of two large epidemiological studies on this population (Nguyen LH et al., Lancet Public Health 2020; Kim H. et al., BMJ Global Health 2021).
  • We agree with the Reviewer that crossreactivity between SARS-CoV-2 and common cold coronaviruses epitopes has been recognized by numerous studies and that low dose exposure with SARS-CoV-2 cannot explain pre-existing immunity in pre-pandemic samples. We acknowledged this fact in the revised manuscript and discussed recent studies on the role of cross-reactive immunity in influencing COVID-19 outcomes.
  • We agree on the importance of viral replication in influencing inflammation and immune response upon SARS-CoV-2 infection. We have added this concept in the section on immune response mechanisms together with new insights on this topic provided by the study of Wang Z. et al. (Nature Communications March 2021, doi 1038/s41467-021-22036-z).
  • We have added to the Conclusions the concept that low-dose exposures to SARS-CoV-2 may be useful in maintaining long-term anti-SARS-CoV-2 immunity post-vaccination.

Round 2

Reviewer 1 Report

The changes to the manuscript type and the presentation are sufficient.

Reviewer 2 Report

      From the previous version of manuscript, I would like to thank the authors for carefully addressing Reviewers’ concerns, especially with regards to the abstract, conclusion/implication and article type revision. This manuscript is also strengthened after adding most updated and relevant publications, which result in a much stronger support of the viewpoint.

      I agree with the authors that the topic of the paper is important, however direct experimental support of the viewpoint is weak and submit as an opinion might be more suitable in this case than a review article. I therefore recommend this paper for publication without further revision.

Reviewer 3 Report

I am happy with the modifications. The authors proved themselves open-minded and received well the feedback.